# The Design and Experimental Validation of a Biomimetic Stubble-Cutting Device Inspired by a Leaf-Cutting Ant’s Mandibles

**DOI:** 10.3390/biomimetics8070555

**Published:** 2023-11-19

**Authors:** Hongyan Qi, Zichao Ma, Zihe Xu, Shuo Wang, Yunhai Ma, Siyang Wu, Mingzhuo Guo

**Affiliations:** 1The College of Biological and Agricultural Engineering, Jilin University, 5988 Renmin Street, Changchun 130025, China; qhy18@mails.jlu.edu.cn (H.Q.); zhxu21@mails.jlu.edu.cn (Z.X.); wangshuol78@163.com (S.W.); 2The Key Laboratory of Bionic Engineering, Ministry of Education, Jilin University, 5988 Renmin Street, Changchun 130025, China; 3Department of Mechanical Engineering, 137 Reber Building, The Pennsylvania State University, University Park, PA 16802-440, USA; zvm5162@psu.edu; 4The College of Engineering and Technology, Jilin Agricultural University, Changchun 130118, China; siyangwu@outlook.com

**Keywords:** leaf-cutting ant, mandible, stubble-cutting device, bionic design, root–soil complex

## Abstract

Under the conditions of conservation tillage, the existence of the root–soil complex greatly increases the resistance and energy consumption of stubble-cutting blades, especially in Northeast China. In this research, the corn root–soil complex in Northeast China was selected as the research object. Based on the multi-toothed structure of the leaf-cutting ant’s mandibles and the unique bite mode of its mandibles on leaves, a gear-tooth, double-disk, bionic stubble-cutting device (BSCD) was developed by using a combination of power cutting and passive cutting. The effects of rotary speed, tillage depth, and forward speed on the torque and power of the BSCD were analyzed using orthogonal tests, and the results showed that all of the factors had a large influence on the torque and power, in the order of tillage depth > rotary speed > forward speed. The performance of the BSCD and the traditional power straight blade (TPSB) was explored using comparative tests. It was found that the optimal stubble-cutting rate of the BSCD was 97.4%. Compared with the TPSB, the torque of the BSCD was reduced by 15.2–16.4%, and the power was reduced by 9.2–11.3%. The excellent performance of the BSCD was due to the multi-toothed structure of the cutting edge and the cutting mode.

## 1. Introduction

Sustainable agricultural development serves as the foundation for sustainable social and economic progress [1,2]. An essential technical component of sustainable agricultural development is conservation tillage, which has gained widespread global adoption and has exhibited favorable economic, social, and ecological outcomes [3,4]. Stubble-cutting devices are essential components of agricultural machinery and can cut off stubble and straw to ensure the effective implementation of conservation tillage. Compared with the traditional tillage mode, conservation tillage puts forward more stringent requirements for stubble-cutting devices in the process of operation [5]. This is because conservation tillage creates a complex farmland working environment mainly composed of crop straw and the root–soil complex, which is formed by crop stubble root and soil. Under the mode of conservation tillage, the agricultural machinery’s working environment has changed from bare soil to farmland incorporating crop straw and the root–soil complex, which results in agricultural machinery needing to overcome greater resistance [6]. Therefore, current stubble-cutting devices necessitate innovative redesign to fulfill the requirements of the root–soil complex farmland operation environment.

The root–soil complex is a composite material formed by the combination of plant root systems and soil. The plant roots within the complex are staggered and tightly bound to the soil, providing stability for the plant and allowing for maximum nutrient and water absorption. Additionally, the roots play a role of stabilization and anchoring in the soil so that the root–soil complex has a more stable structure and higher mechanical strength [7,8,9]. As a result, stubble-cutting devices need to face greater resistance in the process of cutting root–soil complex, which seriously affects the production efficiency and operation quality of agricultural machinery [10].

In the past few decades, scholars have carried out a series of theoretical and practical research on stubble-cutting devices. For instance, Jia et al. devised a gear-tooth cutting mechanism for breaking stubble and optimized the basic parameters, numbers, and edge curves of the cutting blade [11]; Jiang et al. designed a double cutter disc-power-cutting device, and field trials showed that the device can create good seed bed conditions and effectively solve the problem of straw blockage [10]; Zhu et al. adopted the double-eccentric circle method to design a blade edge curve, which improved the plant-crushing rate and reduced power consumption [12]; Quan et al. developed a blade with a multilevel sliding cutting angle and found that optimizing the blade geometry could reduce cutting resistance [13]. In addition, bionics has been shown to be useful in the design of agricultural machinery to reduce resistance and energy consumption [14,15,16]. Researchers have also designed a series of stubble-cutting components using bionics. For example, Zhu et al. extracted the contour curves of the left and right mouths of *Batocera horsfieldi*, and designed a bi-directional rotating blade that can efficiently cut corn stalks [17]; based on the contour structure of a locust’s mouthparts and its distinctive biting technique on maize rootstocks, Zhao et al. developed a bionic stubble-breaking device with a symmetrical rotational motion that can significantly increase the stubble-breaking rate and reduce resistance to stubble-breaking operations [18]; Chang et al. designed two types of biomimetic stubble cutters with different tooth heights by utilizing the front claws of the nymph of *Cryptotympana atrata* and found that the serrated structure design was the main factor in reducing cutting resistance [19].

As indicated above, scholars have made impressive strides in studying stubble-cutting devices. However, previous research has focused primarily on straw and soil cutting, with few exploring stubble-cutting devices in the context of the root–soil complex. Additionally, different regions have varying requirements for operating implements. Northeast China stands as a crucial area for the dissemination and practical application of conservation tillage. Conservation tillage has generated certain advantages in the Northeast region, but it continues to face numerous challenges [20]. The stubble-cutting devices of working machines in Northeast China still have problems with high cutting resistance and torque, as well as high energy consumption, making it difficult to ensure the smooth application of conservation tillage [18,21]. There is an urgent need to devise new stubble-cutting devices according to the operating environment in the Northeast region to meet operating needs.

Leaf-cutting ants are social swarming insects, classified in the genera *Atta* and *Acromyrmex*, and are found mainly in Central and South America, as well as in Mexico and Southern United States [22]. They are known as ecosystem engineers because of their ability to influence soil physical and chemical properties and plant community composition [23,24]. In recent years, leaf-cutting ants have been the focus of scientific research. Researchers have conducted numerous studies on these ants and obtained significant findings [25,26,27]. The most distinctive appearance of the leaf-cutting ant is its large head and pair of mandibles. Its mandibles serve many functions, such as prey capture, fighting, brood rearing, and communication [28]. Besides that, the mandibles are also used for digging nests and cutting leaves, exhibiting excellent cutting performance [29,30]. Studies have shown that the mandibles of leaf-cutting ants have a large biting function, with a bite force 2600 times their own body weight [31]. The mandibles of leaf-cutting ants are able to withstand impacts and loads during the exercise of their functions, showing excellent mechanical properties, and the mandibles also show morphological adaptations in order to realize efficient cutting functions [32]. It is not difficult to find that the mandibles of leaf-cutting ants are natural and excellent bionic blueprints, and their morphology can be used for the bionic design of engineering applications. However, to our knowledge, there is no research on the use of the leaf-cutting ant’s mandibles for the bionic design of engineering applications at this time.

In summary, in view of the current situation of high cutting resistance and high energy consumption required for cutting in the corn root–soil complex in Northeast China, we took the corn root–soil complex in Northeast China as the object of study in this paper. Based on the morphology of the Atta’s mandibles and its biting mode in the process of cutting leaves, we developed a gear-tooth, double-disk, bionic stubble-cutting device (BSCD) by adopting a cutting case that combines power cutting and passive cutting. The effects of operating parameters on the BSCD were analyzed through soil bin tests and field experiments, and the mechanisms of resistance reduction and energy saving of the bionic stubble-cutting tool device was explored. The results of this research are expected to provide a theoretical basis for the cutting tool design of the root–soil complex and are expected to provide technical support for conservation tillage equipment in Northeast China.

## 2. Materials and Methods

### 2.1. Design of the BSCD

The leaf-cutting process of leaf-cutting ants has been extensively studied and described in the existing literature [26,33]. These ants use a supported cutting method with two mandibles to cut leaves asymmetrically at varying speeds, resulting in efficient cutting. The appearance of leaf-cutting ants was macroscopically observed with a stereomicroscope (StereoDiscovery.V12, ZEISS, Jena, Germany) and a microscope (Digitalstereomicroscope.VHX-6000, KEYENCE, Daiba, Japan). The morphology of leaf-cutting ants is shown in Figure 1a,d. The leaf-cutting ant can be observed to have three pairs of ridged, elevated structures on its back (Figure 1d). The most obvious external feature of the leaf-cutting ant is its pair of large mandibles (Figure 1b,c). The mandibles have a multi-toothed structure, which is helpful in cutting leaves.

Based on the bite mode of leaf-cutting ants’ mandibles and the multi-toothed contour structure of the mandibles, we designed a gear-tooth BSCD with a supported double-disk structure. The BSCD is shown in Figure 2. It mainly consists of a power disc base, a power disc, a passive disc base, a passive disc, and bionic stubble-cutting blades (Figure 2a,b). The dimensions of the power disc and the passive disc are the same except for the size of the disc through the hole. The disc base, the disc, and the bionic stubble-cutting blades are fixed by bolts and nuts to form the power bionic stubble-cutting disc and the passive bionic stubble-cutting disc, respectively. As shown in Figure 2c, there are six bionic stubble-cutting blades on each disc. The rotation direction of the passive stubble-cutting disc is consistent with the forward direction of the machine, and it rotates with the advance of the machine. The power stubble-cutting disc is powered by the tractor’s power output shaft and rotates in the opposite direction to the passive stubble-cutting disc (see Figure 2a).

The bionic stubble-cutting blade is designed based on the multi-toothed structure of the leaf-cutting ant’s mandibles; therefore, it is necessary to extract the multi-toothed contour curves of the mandibles. The outline structure of the leaf-cutting ant’s mandibles can be observed in Figure 1c and Figure 3a (The leaf-cutting ant’s mandible in Figure 3a was selected from twenty-five samples of the sharpest). MATLAB and Origin software are used to extract and fit the outer edge contour curves of the mandible. Firstly, the rgb2gray, imerode, imdilate, im2bw, Imfill, and edge function commands in Matlab software were individually used to process the profilogram of Figure 3a; therefore, it was converted from an original image to a binary image, and curve contour coordinate points were derived. Then, the LOG algorithm was used to plot the coordinate points into the final boundary map, and the profilogram was divided into seven independent curves and a straight line perpendicular to the *X*-axis (see Figure 3b). Finally, the seven outer edge contour curves were fitted using Origin software and the Polynomial Fit command to obtain fitted curves (Figure 3c). When comparing the outer edge contour curves and the fitted curves, it is possible to find that they almost completely overlap (see Figure 3d).

The least-squares method was applied to obtain Equation (1) for the fitting of the cutting-edge curve of the leaf-cutting ant’s mandibles, and the equation is as follows:(1)$$\varphi \left(x\right)=B_{0}+B_{1}x^{1}+B_{2}x^{2}+B_{3}x^{3}+B_{4}x^{4}+B_{5}x^{5}+B_{6}x^{6}+B_{7}x^{7}+B_{8}x^{8}+B_{9}x^{9}$$

In order to ensure that the format of the fitting results for each curve is relatively uniform, and to facilitate the parameter comparison and engineering application of each curve, the *X*-axis coordinates of these seven curves were initialized (Figure 4). The parameter values of *B*_0_–*B*_9_ were obtained, as shown in Table 1 and Table 2. It is not difficult to find that the fitting variance R^2^ is greater than 0.99, which indicates that the accuracy of the fit curves is high.

Through the curve equations above, the fit curves were scaled in proportion by using AutoCAD 2018 and SOLIDWORKS 2018 software, which was used to design the cutting edge of the bionic stubble-cutting blade. The multi-tooth contour curve of the mandible arrayed twice was used to design the cutting edge of the bionic stubble-cutting blade, and the actual object of the bionic stubble-cutting blade was processed using laser-cutting technology.

### 2.2. Soil Bin Tests

The stubble-cutting tests of the root–soil complex were carried out in the indoor soil bin laboratory (40 m long and 3 m wide) of Jilin University. The corn root–soil complex (Figure 5c,d) used in the experiment was collected from the agricultural experimental base of Jilin University, and the samples were wrapped in cling film after collection to prevent water loss. In order to simulate the field experimental environment, the soil in the soil bin needed to be rectified (see Figure 5). The soil preparation technology that was adopted was as follows: soil rotary tillage (Figure 5a); watering; soil leveling; compaction (Figure 5b); root–soil complex burial (Figure 5e); and compaction, and the process was not completed until the soil conditions were similar to those in the field. The soil bulk density of 1.366–1.373 g/cm^3^ and soil moisture content of 19.71–19.83% were measured through the oven drying method, and the soil cone index of 1.000–1.035 MPa was measured using the SC-900 type Soil Compactor Meter (RGB Spectrum Equipment, Alameda, CA, USA) with a 1/2″ 00 diameter cone tip.

The test equipment consisted of a soil bin tester system, a power transmission system (single row operation), and a BSCD, as shown in Figure 6. The soil bin tester system mainly included a soil bin testing trolley and data acquisition sensors (torque sensor, CYB-803S, Weste Aviation Technology Co., Ltd. (Shenzhen, China); upper pull rod sensor (BK-1-LG) and lower suspension pin sensor (BK-5-XG), China Academy of Aerospace Aerodynamics, Beijing, China). The power transmission system and the BSCD were developed by us (Figure 2a and Figure 6b,e). The power transmission system included the frame, gearbox (transmission ratio 3:1, see Figure 6d), sprocket transmission (transmission ratio 1:1, see Figure 6c), and drive shafts.

Figure 7 illustrates the soil bin test area. The soil bin test area was divided into three sections. The two ends were set up as two transition sections, each with a total length of 5 m. The middle section was selected as the stable section for recording the test data, with a total length of 6 m. Three corn root–soil complexes were taken and buried in the stable section, with a spacing of 1 m between each of the corn root–soil complexes. The distance between corn plants in the field was between 0.2–0.4 m, and the lateral length of the corn root–soil complex was about 0.3 m. When the machine was in operation, one corn root–soil complex was cut every 0.3 m on average. Considering the data above and in order to simulate the field test environment as much as possible, the data within the 0.3 m area before and after the cutting of the corn root–soil complex in the soil tank test were selected as the test data for analysis.

Cutting torque and power were selected as the test evaluation indexes for the soil bin test. To optimize the operating parameters and investigate the effects of operating condition parameters (forward speed, rotary speed, and tillage depth) on cutting torque and power, the orthogonal combination test of three factors and three levels was designed by using the Box–Behnken Design principle of the Design-Expert software. Table 3 presents the factor codes. The experimental program comprised 17 groups of tests, including 12 groups of factorial point tests and 5 groups of zero-point tests. Each group of tests was repeated three times, and the average value was obtained in order to analyze the results. The significant effects of the factors on the test indicators were examined to obtain the response surface and regression equation.

To investigate the performance of the BSCD designed, the traditional power straight blade (TPSB) was selected to compare cutting torque and power. The TPSB consists of a power disc and six straight blades. The size of the straight blade and the bionic stubble-cutting blade are the same; the difference is in the shape of the cutting edge (see Figure 8a,b). During the comparison test, the soil bin testing trolley maintained a forward speed of 3 km·h^−1^ while the rotary speed was varied between 120 rpm and 240 rpm every 60 revolutions. The tests were repeated three times per set, and the average value of the results was used for comparison and discussion.

Cutting torque was calculated as shown in Equation (2), and power was calculated with reference to Equations (3)–(5),
(2)$$T=T_{1}-T_{0}$$
where T is the cutting torque, N·m; *T*_1_ is the cutting torque when the machine drove the stubble-cutting blade to cut the root–soil complex, N·m; *T*_0_ is the cutting torque measured by the torque sensor when the machine was unloaded, N·m.
(3)$$P_{1}=\frac{2\pi \times \frac{n}{60}\times T}{1000}=\frac{T\times n}{9550}$$
(4)$$P_{2}=\frac{(F_{m}-F_{n})\times V_{m}}{1000}$$
(5)$$P_{3}=P_{1}+P_{2}$$
where *P*_1_ is the operating power of the power stubble-cutting disc; *P*_2_ is the operating power of the passive stubble-cutting disc; *P*_3_ is the operating power of the BSCD, kW; *n* is the rotary speed, rpm; *T* is the cutting torque, N·m; *F_m_* is the horizontal traction resistance of the BSCD when it was in operation, N; *F_n_* is the horizontal traction resistance of the power bionic stubble-cutting disc when it was in operation, N; *V_m_* is the forward speed of the machine, m·s^−1^. Note: The power of the TPSB was obtained using Equation (3), and the power of the BSCD was obtained using Equation (5).

### 2.3. Field Experiments

The field experiments were conducted at the agricultural experimental base of Jilin University in Changchun, Jilin Province, China (43°57′4″ N,125°14′52″ E). The previous crop was corn, and the plant row spacing was 650 mm. Before the beginning of the experiment, the soil bulk density and soil moisture content were measured through the oven drying method, and the values of the soil bulk density and soil volumetric moisture content were 1.37 g/cm^3^ and 19.77%, respectively. The soil cone index of 1.035 MPa was measured using the SC-900 type Soil Compactor Meter (RGB Spectrum Equipment, USA) with a 1/2″ 00 diameter cone tip.

Figure 9 shows the experimental equipment in the field. The field test was carried out by a tractor (John Deere B554, Figure 9a) driving the stubble-cutting device (Figure 9c,d) through the power transmission system (dual-row operation, Figure 9b) to cut the root–soil complex. The test data were measured by the agricultural machinery dynamic parameter remote-sensing instrument and the torque sensor of the Jilin Agricultural University.

The evaluation indexes of the field test are cutting torque, power, and stubble-cutting rate. The cutting torque calculation method is shown in Equation (6), power was calculated using Equations (3)–(5), and the stubble-cutting rate was calculated using Equation (7).
(6)$$T=\frac{T_{1}-T_{0}}{n}$$
where *T* is the cutting torque, N·m; *T*_1_ is the cutting torque when the machine drove the stubble-cutting blade to cut the root–soil complex, N·m; *T*_0_ is the cutting torque measured by the torque sensor when the machine was unloaded, N·m; *n* is the number of job rows in this experiment, *n* = 2.
(7)$$y=\frac{\sum_{i=1}^{3}\frac{n_{i}}{n_{i1}}}{3}\text{×}100\%$$
where *n*_i1_ is the total number of root stubble before machine operation in each data collection area; *n*_i_ is the total number of root stubble cut and broken after machine operation in each data collection area.

Comparative tests were designed to compare the operational performance of the BSCD with the TPSB. The rotary speed was kept at 240 rpm to compare their operational performance at different tractor forward speeds (3 km·h^−1^, 4 km·h^−1^ and 5 km·h^−1^). The test length of each set of tests was 70 m, including a 10 m long transition section at both ends and a 50 m long stable section in the middle part. Each set of tests was repeated three times, and the average value of the test data was taken for analysis.

## 3. Results and Discussion

### 3.1. Results of Soil Bin Tests

The design scheme and results of the orthogonal combination test are shown in Table 4. By comparing the test data and observing the fluctuation changes in cutting torque and power, we can obtain the preliminary finding that both cutting torque and power increase with the increase in tillage depth and forward speed. The cutting torque decreases with the increase in forward speed, while the power increases with the increase in forward speed. Using Design-Expert to process the results of the test data in Table 4, we were able to obtain the results of the analysis of variance for cutting torque and power (Table 5 and Table 6, respectively).

Quadratic multiple regression model equations for the coded independent variables of cutting torque and power consumption were established by fitting the data in Table 5 and Table 6 using the least squares method. The coded independent variable quadratic multiple regression model equation for cutting torque is shown in Equation (8) below, and the coded independent variable regression model equation for power consumption is shown in Equation (9) below.
(8)$$\begin{array}{l} Y_{1}=33.4-6.18X_{1}+12.23X_{2}+3.66X_{3}-0.7825X_{1}X_{2}-0.175X_{1}X_{3}+0.9175X_{2}X_{3}+0.2305X_{1}^{2}-2.7X_{2}^{2}\\ +0.2855X_{3}^{2} \end{array}$$
(9)$$\begin{array}{l} Y_{2}=0.6635+0.0869X_{1}+0.2499X_{2}+0.081X_{3}+0.0645X_{1}X_{2}+0.0192X_{1}X_{3}+0.0236X_{2}X_{3}-0.0345X_{1}^{2}\\ -0.0423X_{2}^{2}+0.0047X_{3}^{2} \end{array}$$
where *Y*_1_ is the cutting torque, N·m; *Y*_2_ represents the power, kW.

The reliability of the cutting torque regression model equation was analyzed in conjunction with Table 5. It was able to find *p* < 0.0001 for the model, which means that the model is reasonable and significant. The model factors X_1_, X_2_, X_3_, X_1 × 2_, X_2 × 3_, and X_2_^2^ are significant terms (*p* < 0.05), and the rest of the terms are insignificant. The *p*-value of the lack of fit is 0.2696, which is insignificant and indicates that the quadratic multiple regression model equation is acceptable.

It can be observed from Table 6 that the *p* value of the model is less than 0.0001, and the *p* value of the misfit term is insignificant, which indicates that the regression equation of power is reasonable. The *p*-value of X_1_, X_2_, X_3_, X_1 × 2_, X_1 × 3_, X_2 × 3_, X_1_^2^, and X_2_^2^ are less than 0.05, and they are significant terms, whereas the *p*-value of X_3_^2^ is more than 0.05, and it is an insignificant term.

The response surface method was applied to analyze the effects of three influencing factors (rotary speed, tillage depth, and forward speed) and their interactions on the test indexes (cutting torque and power). During the analysis, one factor was fixed at zero level, and the effect of the remaining two factors was then analyzed and discussed.

When analyzing the effects of the three factors and their interactions on the cutting torque, the effects of tillage depth and rotary speed on the cutting torque at a fixed forward speed of 4 km·h^−1^ can be expressed as follows:(10)$$Y_{1}=33.4-6.18X_{1}+12.23X_{2}-0.7825X_{1}X_{2}+0.2305X_{1}^{2}-2.7X_{2}^{2}$$

When the tillage depth is fixed at 65 mm, the influence of forward speed and rotary speed on cutting torque can be expressed as follows:(11)$$Y_{1}=33.4-6.18X_{1}+3.66X_{3}-0.175X_{1}X_{3}+0.2305X_{1}^{2}+0.2855X_{3}^{2}$$

When the rotary speed is fixed at 180 rpm, the influence of tillage depth and forward speed on cutting torque is as follows:(12)$$Y_{1}=33.4+12.23X_{2}+3.66X_{3}+0.9175X_{2}X_{3}-2.7X_{2}^{2}+0.2855X_{3}^{2}$$

Figure 10 is the response surface of the influence of the three factors on cutting torque. Combining Table 5, Figure 10, and Equations (10)–(12), it can be seen that the three factors have a significant impact on the cutting torque. The tillage depth has an interaction with the forward speed and the rotary speed, respectively, but there is no interaction between the rotary speed and the forward speed. The cutting torque increases with the increase in tillage depth and forward speed and decreases with the increase in rotary speed. By observing the changing trend of cutting torque with the three factors in Figure 10, it can be found from Figure 10a that the change in cutting torque from “−1” to “1” with the level of factor D is greater than that from “−1” to “1” with the level of factor n, which shows the order of influence on cutting torque: tillage depth > rotary speed. Similarly, the order of influence on cutting torque can be found in Figure 10b: rotary speed > forward speed. The order of influence on cutting torque can be found from Figure 10c: tillage depth > forward speed. To summarize, the influence order of the three factors on cutting torque is as follows: tillage depth > rotary speed > forward speed.

When analyzing the impact of the three factors and their interaction on power, with a fixed forward speed of 4 km·h^−1^, the impact of tillage depth and rotary speed on power can be expressed as follows:(13)$$Y_{2}=0.6635+0.0869X_{1}+0.2499X_{2}+0.0645X_{1}X_{2}-0.0345X_{1}^{2}-0.0423X_{2}^{2}$$

The effect of forward speed and rotary speed on power at a fixed tillage depth of 65 mm can be expressed as follows:(14)$$Y_{2}=0.6635+0.0869X_{1}+0.081X_{3}+0.0192X_{1}X_{3}-0.0345X_{1}^{2}+0.0047X_{3}^{2}$$

The effect of tillage depth and forward speed on power at a fixed rotary speed of 180 rpm is as follows:(15)$$Y_{2}=0.6635+0.2499X_{2}+0.081X_{3}+0.0236X_{2}X_{3}-0.0423X_{2}^{2}+0.0047X_{3}^{2}$$

Observing the response surface plots of the effects of the three factors on power (Figure 11), combined with Table 6 and Equations (13)–(15), each factor can be demonstrated to have a significant effect on power. There are interactions between each of the three factors. With the increase in the three factors, the power increases. Observing the changing trend of power with the three factors in Figure 11, it can be found from Figure 11a that the amount of change in power with the level of factor D from “−1” to “1” is greater than the amount of change in power with the level of factor n from “−1” to “1”, which indicates the order of influence on power: tillage depth > rotary speed. Similarly, the order of influence on power can be found from Figure 11b: rotary speed > forward speed. From Figure 11c, the order of influence on power can be found as follows: tillage depth > forward speed. Therefore, the order of influence of the three factors on power is tillage depth > rotary speed > forward speed.

Using the regression equation of torque and power and the Design-expert software, the optimal solution for the operating parameters under the conditions of minimum torque and minimum power was found. Considering the actual size of the root–soil complex and in order to ensure the stubble-cutting effect, we choose the following operating parameters for practical application: a rotary speed of 240 rpm, a tillage depth of 90 mm, and a forward speed of 3 km·h^−1^.

The results of the soil bin comparison tests are shown in Figure 12. From Figure 12a,b, we are able to observe that the torque of the BSCD and the TPSB decreases with the increase in rotary speed, while the power increases with the increase in rotary speed, which is similar to the research results of Matin and Yang et al. [34,35]. At the rotary speed of 240 rpm, the torque is the smallest, while the power is the largest. The torque of the BSCD and the TPSB are 31.89 N m and 37.85 N m, and the power is 0.844 kW and 0.951 Kw, respectively. The torque and power of the BSCD is always less than that of the TPSB. Under the three working conditions of 120 rpm, 180 rpm, and 240 rpm, the cutting torque of the BSCD is reduced by 16.4%, 15.2%, and 15.7%, respectively, and power consumption is reduced by 9.9%, 10.1%, and 11.3%, respectively, compared with the TPSB. This indicates that the BSCD has a better resistance reduction and energy-saving performance.

### 3.2. Results of Field Experiments

The cutting torque results of the field comparison tests can be observed in Figure 13. It is not difficult to find that the changing trends of the BSCD and the TPSB are similar, and their cutting torque gradually increases with the increase in forward speed. This change is consistent with the results of previous studies [18,36,37]. The reason for this changing trend is because the bite length of the blades increases with the increase in forward speed, resulting in an increase in cutting torque. The torque of the BSCD is always lower than that of the TPSB under operating conditions of 3 km·h^−1^, 4 km·h^−1^, and 5 km·h^−1^. Compared with the TPSB, the cutting torque of the BSCD is reduced by 15.4%, 15.8%, and 16.1%, respectively, which means that the bionic cutting tool device has better drag reduction.

Figure 14 shows the power results of the BSCD and TPSB in the field comparison tests. The power is, at minimum, 3 km·h^−1^ and, at maximum, 5 km·h^−1^. The trend of power at different forward speeds is similar to the trend of power at different rotary speeds in the soil bin test. The power of the TPSB is always greater than that of the BSCD for the different forward speed operating conditions (3 km·h^−1^, 4 km·h^−1^ and 5 km·h^−1^), and the power of the BSCD is reduced by 11%, 10.2%, and 9.2%, respectively, compared to the TPSB. This indicates that the BSCD has better energy-saving performance.

The results of the stubble-cutting rate of the BSCD and TPSB in the field comparison test are shown in Figure 15. The stubble-cutting rate decreases with increasing speed. This is because, at a given rotational speed and unit cutting length, the greater the forward speed, the fewer times the blade cuts the stubble, and the cutting effect becomes worse. Under the operating conditions of 3 km/h, the stubble-cutting rate of the BSCD reached 97.2%, and the stubble-cutting rate of the TPSB was 93.4%. The stubble-cutting rate of the BSCD is always greater than that of the TPSB during the test, which indicates that the BSCD has a better stubble-cutting performance.

### 3.3. Discussion on the Mechanism of Drag Reduction and Energy Saving of the BSCD

The results of the above-mentioned soil tank comparison test and field comparison test have proven that the BSCD has a better operating performance than the TPSB under different operating conditions of rotary speed and forward speed. The increased performance of the BSCD reflects two factors: the cutting-edge structure of the blade and the way the blades cut the root–soil complex.

The cutting edge of the BSCD is a designed curve based on the multi-toothed contour structure of the leaf-cutting ant’s mandibles, while the cutting edge of the TPSB is a smooth straight line. When they cut the root–soil complex at the same depth, the effective contact area between the cutting edge of the BSCD and the root–soil complex is smaller than that between the cutting edge of the TPSB and the root–soil complex, and the length of the bionic cutting edge is larger than that of the ordinary cutting edge, which results in the resistance of the root–soil complex to the BSCD, being less than the resistance to the TPSB (see Figure 16a,b). The cutting edge of the BSCD is multi-toothed. Compared with the cutting edge of the TPSB, the multi-toothed cutting edge has stronger penetration performance [38]. When the BSCD cuts the root–soil complex, the tooth-shaped structure penetrates the root–soil complex, which can cause local damage to the penetrated part and produce cracks, thereby reducing the overall strength of the root–soil complex and reducing cutting resistance. In addition, the outer contour curve of each tooth-shaped structure has sliding/cutting performance, which can further reduce resistance to cutting the root–soil complex.

The BSCD is a double disc cutting device composed of power cutting blades and passive cutting blades. When the BSCD cuts the root–soil complex, the movement direction of the power cutting blades and the passive cutting blades is opposite to each other, and the cutting speed of the power cutting blades is much higher than the cutting speed of the passive cutting blades, which can produce a large relative motion speed between them. On the basis of the multi-toothed structure piercing the root–soil complex, the power cutting blades and the passive cutting blades rely on relative motion speed to shear the root–soil complex, which reduces the slip distance of the root–soil complex and greatly reduces cutting torque and power consumption (Figure 17a). The cutting edge of the TPSB is a smooth straight line without a piercing and slip cutting effect. It cuts the root–soil complex in the form of smooth chopping, and the root–soil complex may slip when being cut, requiring greater torque and energy consumption to complete the cutting operation (Figure 17b).

In summary, the excellent performance of the BSCD is due to its cutting-edge structure and operational cutting mode. Compared with the TPSB, the BSCD has better resistance reduction and energy-saving performance.

## 4. Conclusions

Based on the multi-toothed contour curve structure and bite mode of the leaf-cutting ants’ mandibles, a gear-tooth, double-disk, bionic stubble-cutting device (BSCD) was developed by using the bionic design method. The results of soil bin tests and field experiments are as follows:(1)The three operating parameters of rotational speed, tillage depth, and forward speed affect cutting torque and power consumption in the following order of magnitude: tillage depth, rotational speed, and forward speed. Taking the minimum torque and power as the target and considering the stubble-breaking effect, the optimal operating parameters determined by combining the response surface method are as follows: rotational speed = 240 rpm, plowing depth = 90 mm, and forward speed = 3 km·h^−1^.(2)The cutting torque and power of the BSCD were consistently lower than those of the TPSB under different RPM and forward speed operating conditions, with a reduction in cutting torque of 15.2–16.4% and a reduction in power of 9.2–11.3%.(3)The BSCD has a very good stubble-cutting performance; its best stubble-cutting rate can reach 97.4%, which is higher than the that of the TPSB.(4)The BSCD offers the advantages of low torque, low power consumption, and a high stubble-cutting rate. Its excellent operating performance is due to the multi-toothed structure of the cutting edge and the cutting mode.

Overall, the geometry and bite mode of leaf-cutting ants’ mandibles play an important role in reducing the torque and energy consumption of stubble-cutting tools. The results of the research in this article can provide theoretical basis and reference for the design of root–soil complex stubble-cutting tools and provide useful inspiration and a new direction for the bionic design and development of agricultural machinery in Northeast China.

## Figures and Tables

**Figure 1 biomimetics-08-00555-f001:**
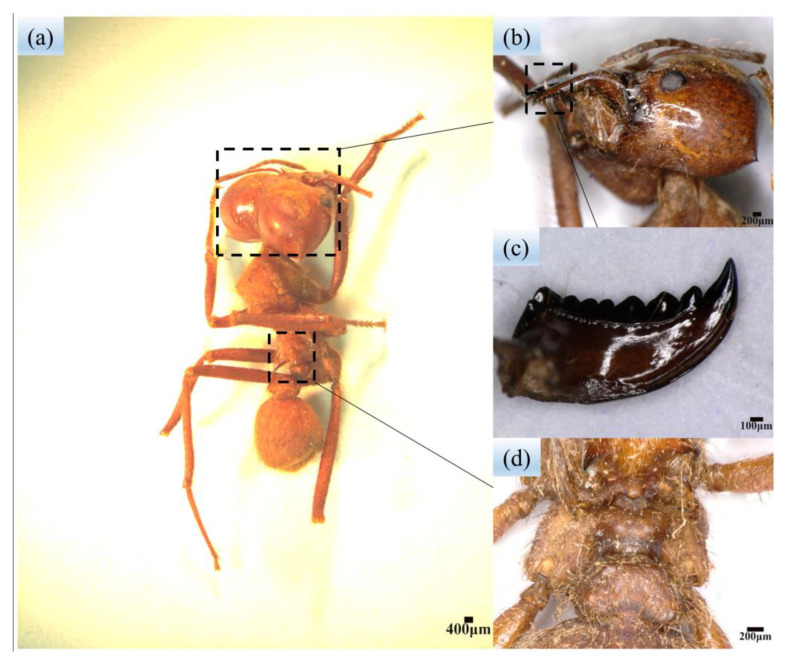
Macroscopic morphology of leaf-cutting ants: (**a**) Body structure of leaf-cutting ants; (**b**,**c**) Mandibles of leaf-cutting ants; (**d**) Ridged elevation on the back of leaf-cutting ants.

**Figure 2 biomimetics-08-00555-f002:**
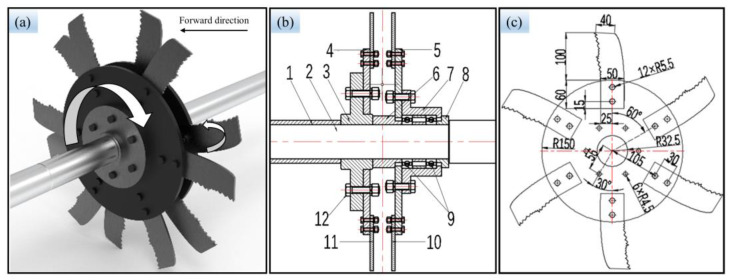
Bionic stubble-cutting device: (**a**) 3D design of the BSCD; (**b**) 2D structure of the BSCD: (1) Shaft sleeve I, (2) Drive shaft, (3) Power disc base, (4) Power disc, (5) Passive disc, (6) Shaft sleeve II, (7) Passive disc base, (8) Shaft sleeve III, (9) Bearings, (10) and (11) Bionic stubble-cutting blades, (12) Bolts and nuts; (**c**) Design dimensions of the power disc and bionic stubble-cutting blades (unit, mm): it is worth noting that the differences between the power disc and passive disc are that the distance 56 changes to 52 mm, and the radius of the center hole R32.5 changes to R28.

**Figure 3 biomimetics-08-00555-f003:**
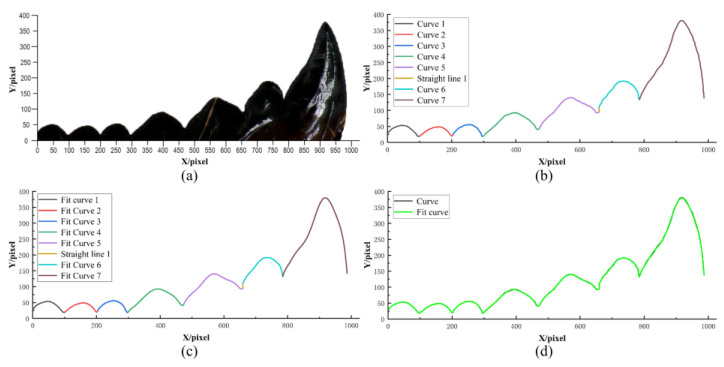
Extraction and fitting of multi-toothed contour curves of the leaf-cutting ant’s mandibles: (**a**) Multi-toothed contour of the leaf-cutting ant’s mandibles; (**b**) Division of multi-toothed contour curves; (**c**) Fitting of multi-toothed contour curves; (**d**) Comparison of multi-toothed contour curves and fitting curves.

**Figure 4 biomimetics-08-00555-f004:**
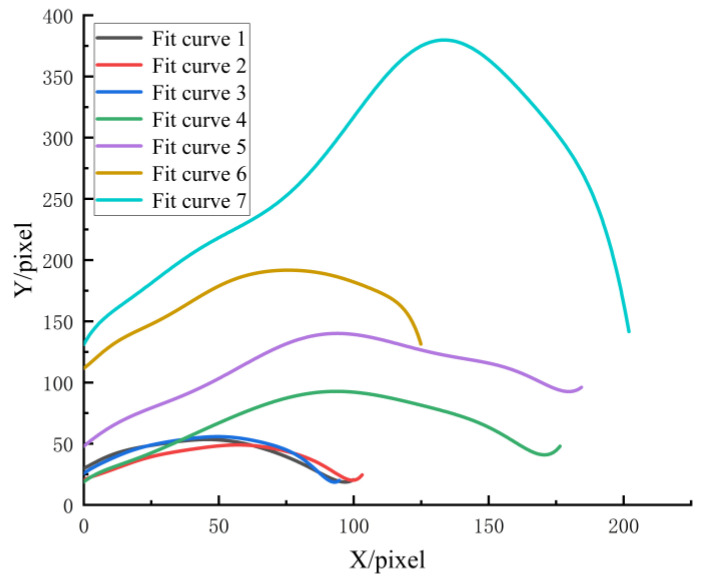
Fit curves for initializing *X*-axis coordinates.

**Figure 5 biomimetics-08-00555-f005:**
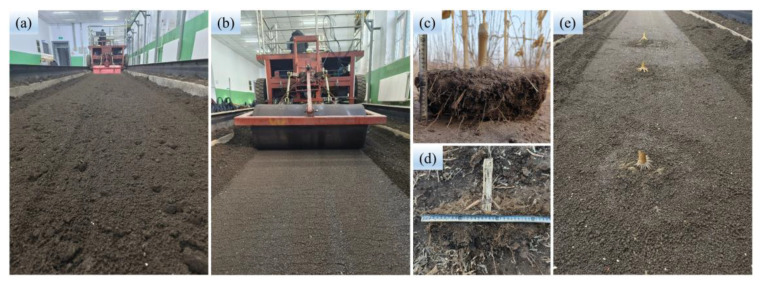
Soil rectification: (**a**) Soil rotary tillage; (**b**) Soil compaction; (**c**,**d**) Root–soil complex collection; (**e**) Root–soil complex burial.

**Figure 6 biomimetics-08-00555-f006:**
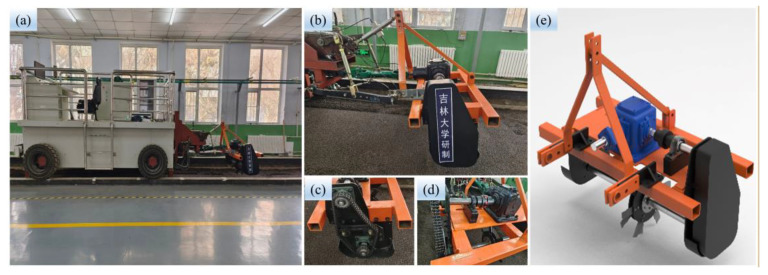
Soil bin test equipment: (**a**) Soil bin tester system; (**b**) Power transmission system; (**c**) Sprocket transmission; (**d**) Gearbox; (**e**) Three-dimensional design drawing of the power transmission system.

**Figure 7 biomimetics-08-00555-f007:**
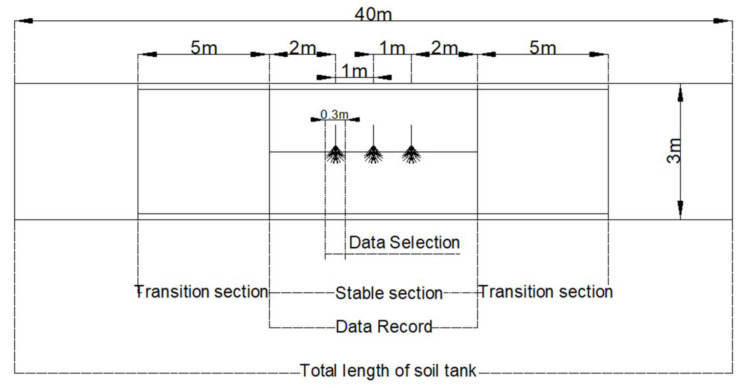
Division of the soil bin test area.

**Figure 8 biomimetics-08-00555-f008:**
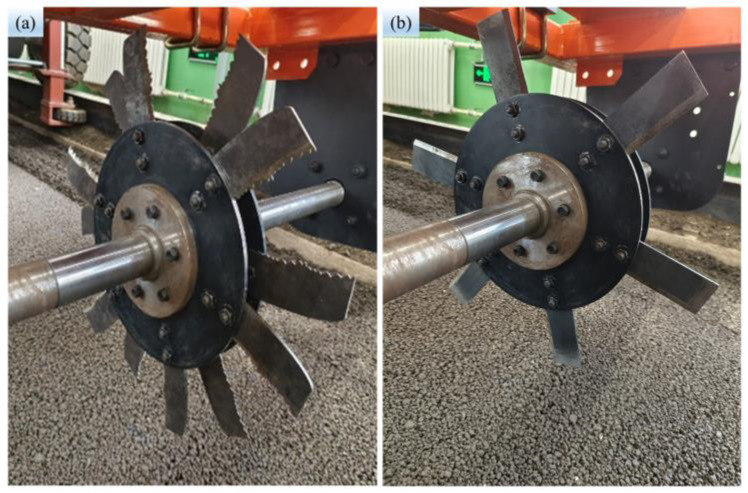
Stubble-cutting blade: (**a**) BSCD; (**b**) TPSB.

**Figure 9 biomimetics-08-00555-f009:**
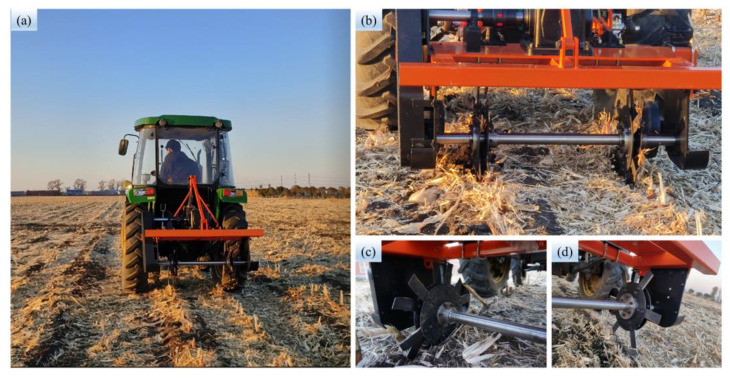
Field test equipment: (**a**) Test tractor; (**b**) Power transmission system; (**c**) BSCD; (**d**) TPSB.

**Figure 10 biomimetics-08-00555-f010:**
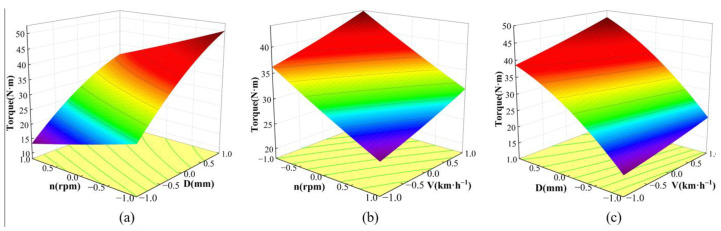
Response surface of cutting torque under the influence of three factors: (**a**) V = 4 km·h^−1^; (**b**) D = 65 mm; (**c**) *n* = 180 rpm. Note: *n* is the rotary speed—the unit is rpm; D is the tillage depth—the unit is mm; V is the forward speed—the unit is km·h^−1^.

**Figure 11 biomimetics-08-00555-f011:**
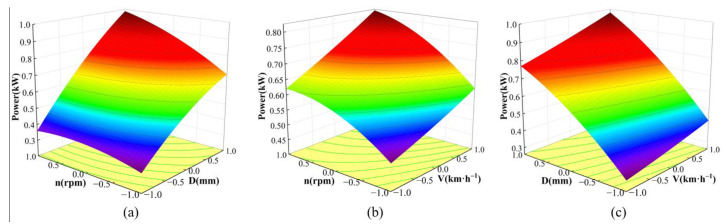
Response surface of power under the influence of the three factors: (**a**) V = 4 km·h^−1^; (**b**) D = 65 mm; (**c**) *n* = 180 rpm.

**Figure 12 biomimetics-08-00555-f012:**
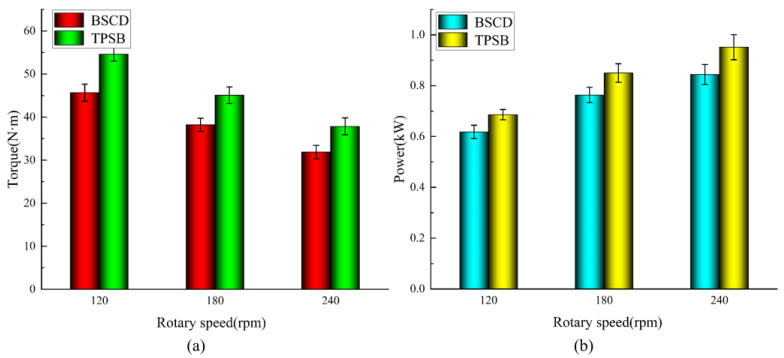
Results of the soil bin comparison tests: (**a**) Effect of rotary speed on the torque of the BSCD and the TPSB; (**b**) Effect of rotary speed on the power of the BSCD and the TPSB. Note: the error bars are standard deviations.

**Figure 13 biomimetics-08-00555-f013:**
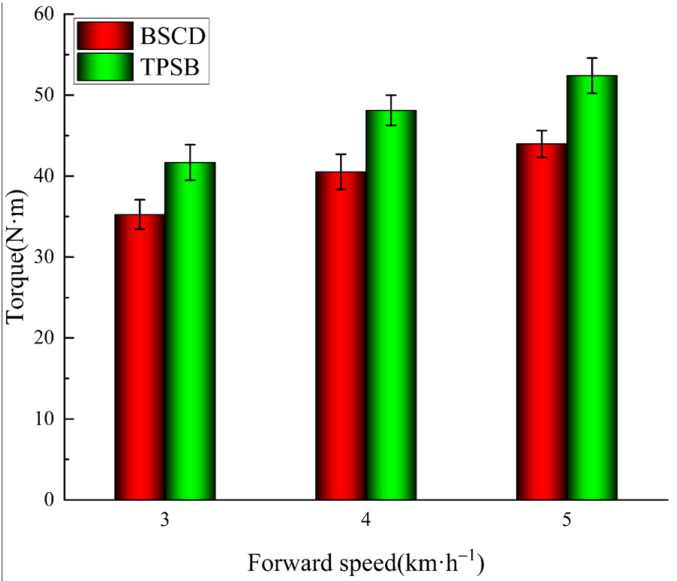
Torque of the BSCD and TPSB at different forward speeds. Note: the error bars are standard deviations.

**Figure 14 biomimetics-08-00555-f014:**
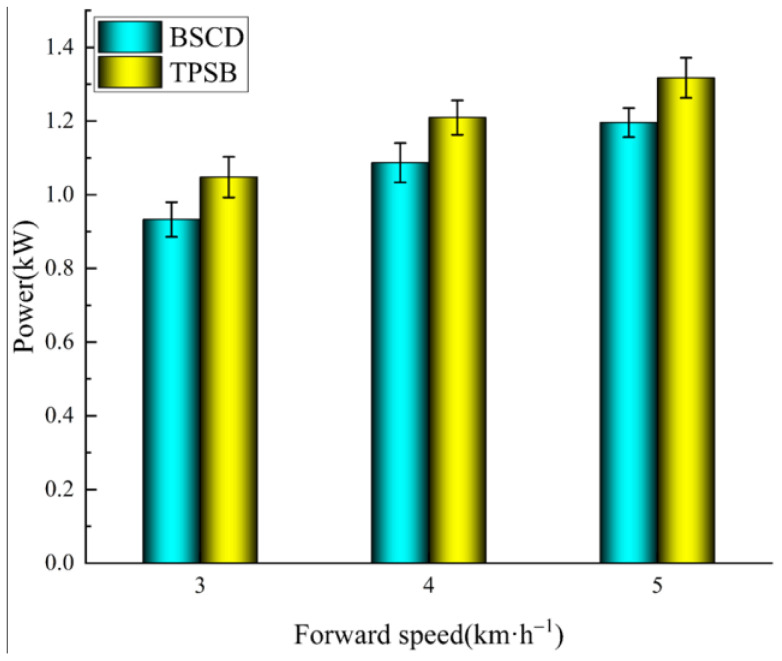
Power of the BSCD and TPSB at different forward speeds. Note: the error bars are standard deviations.

**Figure 15 biomimetics-08-00555-f015:**
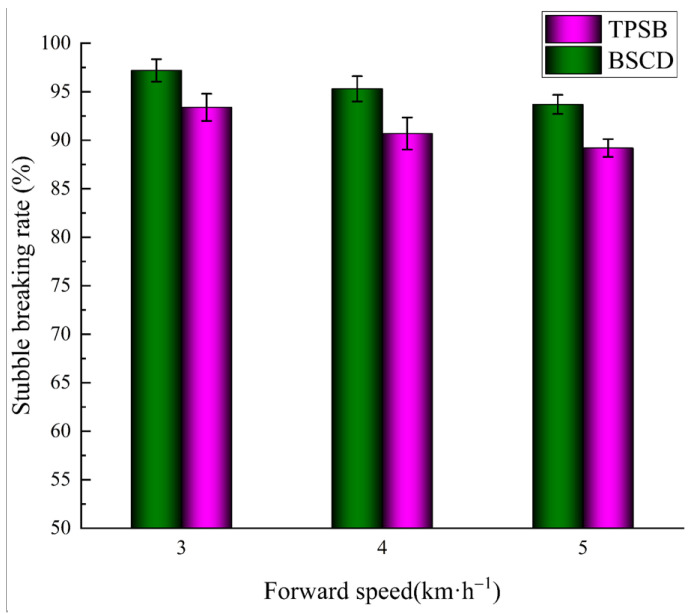
Stubble-cutting rate of the BSCD and TPSB at different forward speeds. Note: the error bars are standard deviations.

**Figure 16 biomimetics-08-00555-f016:**
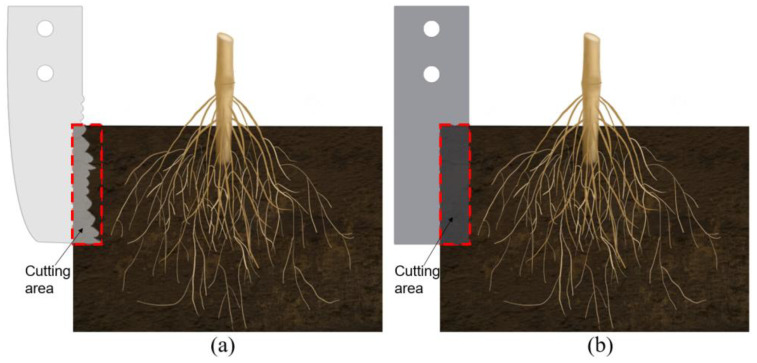
Comparison of the effects of the cutting edges of the BSCD and TPSB on the root–soil complex: (**a**) How the cutting edge of the BSCD cuts the root–soil complex; (**b**) How the cutting edge of the TPSB cuts the root–soil complex.

**Figure 17 biomimetics-08-00555-f017:**
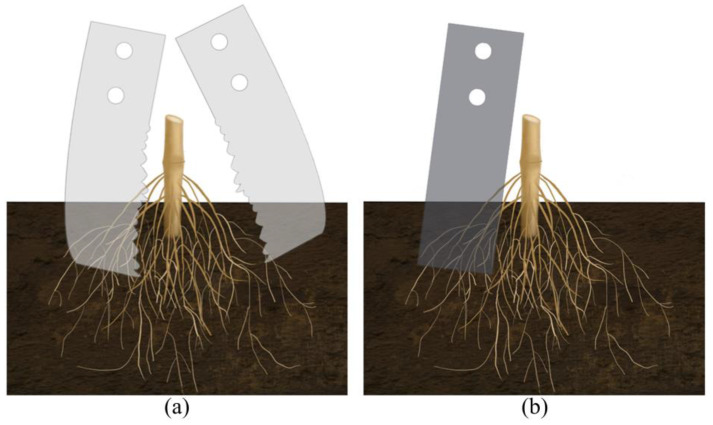
BSCD and TPSB cutting modes in the root–soil complex: (**a**) BSCD cutting mode in the root–soil complex; (**b**) TPSB cutting mode in the root–soil complex.

**Table 1 biomimetics-08-00555-t001:** Curve parameter fitting, Results 1.

	Fit Curve 1	Fit Curve 2	Fit Curve 3	Fit Curve 4
*B* _0_	29.80963	20.32147	25.72501	18.81605
Standard error of *B*_0_	0.24873	0.28949	0.3941	0.62633
*B* _1_	1.16316	1.1197	1.67476	1.73402
Standard error of *B*_1_	0.12915	0.17707	0.27805	0.22124
*B* _2_	0.0219	−0.09234	−0.11186	−0.09646
Standard error of *B*_2_	0.02069	0.03391	0.05962	0.0246
*B* _3_	−0.00488	0.01058	0.01124	0.00479
Standard error of *B*_3_	0.00145	0.00289	0.00566	0.00123
*B* _4_	2.373 × 10^−4^	−6.02588 × 10^−4^	−6.87493 × 10^−4^	−1.28623 × 10^−4^
Standard error of *B*_4_	5.30277 × 10^−5^	1.31047 × 10^−4^	2.85288 × 10^−4^	3.29263 × 10^−5^
*B* _5_	−5.67402 × 10^−6^	1.86173 × 10^−5^	2.37222 × 10^−5^	2.08392 × 10^−6^
Standard error of *B*_5_	1.08574 × 10^−6^	3.45754 × 10^−6^	8.34036 × 10^−6^	5.13266 × 10^−7^
*B* _6_	7.24354 × 10^−8^	−3.32536 × 10^−7^	−4.7649 × 10^−7^	−2.09836 × 10^−8^
Standard error of *B*_6_	1.25466 × 10^−8^	5.46941 × 10^−8^	1.45887 × 10^−7^	4.79828 × 10^−9^
*B* _7_	−4.75336 × 10^−10^	3.43393 × 10^−9^	5.52667 × 10^−9^	1.26857 × 10^−10^
Standard error of *B*_7_	7.63996 × 10^−11^	5.10951 × 10^−10^	1.50449 × 10^−9^	2.64823 × 10^−11^
*B* _8_	1.26393 × 10^−12^	−1.90511 × 10^−11^	−3.43133 × 10^−11^	−4.18508 × 10^−13^
Standard error of *B*_8_	1.90507 × 10^−13^	2.59721 × 10^−12^	8.4306 × 10^−12^	7.94833 × 10^−14^
*B* _9_	0	4.39715 × 10^−14^	8.82263 × 10^−14^	5.76743 × 10^−16^
Standard error of *B*_9_	0	5.53552 × 10^−15^	1.9786 × 10^−14^	9.99645 × 10^−17^
R^2^	0.99921	0.99894	0.99808	0.99855

**Table 2 biomimetics-08-00555-t002:** Curve parameter fitting, Results 2.

	Fit Curve 5	Fit Curve 6	Fit Curve 7
*B* _0_	48.024	111.74722	131.17541
Standard error of *B*_0_	0.48739	1.0711	1.6702
*B* _1_	1.81779	1.56876	4.07523
Standard error of *B*_1_	0.17141	0.55896	0.52096
*B* _2_	−0.01905	0.11275	−0.24141
Standard error of *B*_2_	0.01879	0.09355	0.05035
*B* _3_	−8.47966 × 10^−4^	−0.01391	0.01172
Standard error of *B*_3_	9.17851 × 10^−4^	0.00682	0.00218
*B* _4_	4.11696 × 10^−5^	6.51882 × 10^−4^	−3.06349 × 10^−4^
Standard error of *B*_4_	2.38711 × 10^−5^	2.61758 × 10^−4^	5.02728 × 10^−5^
*B* _5_	−6.22106 × 10^−7^	−1.59409 × 10^−5^	4.47507 × 10^−6^
Standard error of *B*_5_	3.60044 × 10^−7^	5.79118 × 10^−6^	6.74323 × 10^−7^
*B* _6_	3.94614 × 10^−9^	2.22628 × 10^−7^	−3.73378 × 10^−8^
Standard error of *B*_6_	3.2496 × 10^−9^	7.65013 × 10^−8^	5.42557 × 10^−9^
*B* _7_	−7.48538 × 10^−12^	−1.7975 × 10^−9^	1.76242 × 10^−10^
Standard error of *B*_7_	1.72905 × 10^−11^	5.95384 × 10^−10^	2.58004 × 10^−11^
*B* _8_	−2.27969 × 10^−14^	7.83116 × 10^−12^	−4.37193 × 10^−13^
Standard error of *B*_8_	4.99782 × 10^−14^	2.51753 × 10^−12^	6.68139 × 10^−14^
*B* _9_	8.24005 × 10^−17^	−1.42764 × 10^−14^	4.41765 × 10^−16^
Standard error of *B*_9_	6.04847 × 10^−17^	4.45922 × 10^−15^	7.26076 × 10^−17^
R^2^	0.99934	0.99655	0.99923

**Table 3 biomimetics-08-00555-t003:** Factor codes and their values.

Factor	Code	Level		
		−1	0	1
n	X1	120	180	240
D	X2	40	65	90
V	X3	3	4	5

Note: n is the rotary speed—the unit is rpm; D is the tillage depth—the unit is mm; V is the forward speed—the unit is km·h^−1^.

**Table 4 biomimetics-08-00555-t004:** Design scheme and results of orthogonal combination tests.

Number	n (rpm)	D (mm)	V (km/h)	X_1_	X_2_	X_3_	T (N·m)	P (kW)
1	120	40	4	−1	−1	0	23.47	0.314192
2	240	40	4	1	−1	0	13.16	0.349171
3	120	90	4	−1	1	0	50.27	0.695046
4	240	90	4	1	1	0	36.83	0.988208
5	120	65	3	−1	0	−1	36.54	0.482582
6	240	65	3	1	0	−1	24.06	0.627857
7	120	65	5	−1	0	1	44.13	0.601189
8	240	65	5	1	0	1	30.95	0.823296
9	180	40	3	0	−1	−1	16.36	0.321051
10	180	90	3	0	1	−1	38.21	0.763479
11	180	40	5	0	−1	1	21.93	0.440974
12	180	90	5	0	1	1	47.45	0.977871
13	180	65	4	0	0	0	34.19	0.677952
14	180	65	4	0	0	0	32.69	0.65069
15	180	65	4	0	0	0	33.5	0.665091
16	180	65	4	0	0	0	33.41	0.663805
17	180	65	4	0	0	0	33.23	0.659791

**Table 5 biomimetics-08-00555-t005:** ANOVA table of regression model on cutting torque.

Source of Variation	Sum of Squares	Degree of Freedom	Mean Squares	F	*p*
Model	1645.79	9	182.87	451.04	<0.0001
X_1_	305.17	1	305.17	752.71	<0.0001
X_2_	1196.58	1	1196.58	2951.41	<0.0001
X_3_	107.24	1	107.24	264.51	<0.0001
X_1_ X_2_	2.45	1	2.45	6.04	0.0436
X_1_ X_3_	0.1225	1	0.1225	0.3021	0.5996
X_2_ X_3_	3.37	1	3.37	8.31	0.0236
X_1_^2^	0.2237	1	0.2237	0.5518	0.4818
X_2_^2^	30.74	1	30.74	75.82	<0.0001
X_3_^2^	0.3432	1	0.3432	0.8465	0.3881
Residual	2.84	7	0.4054		
Lack of fit	1.67	3	0.5570	1.91	0.2696
Pure error	1.17	4	0.2918		
Cor.Total	1648.63	16			

**Table 6 biomimetics-08-00555-t006:** ANOVA table for regression model on power.

Source of Variation	Sum of Squares	Degree of Freedom	Mean Squares	F	*p*
Model	0.6462	9	0.0718	599.18	<0.0001
X_1_	0.0605	1	0.0605	504.60	<0.0001
X_2_	0.4996	1	0.4996	4169.11	<0.0001
X_3_	0.0525	1	0.0525	438.49	<0.0001
X_1_ X_2_	0.0167	1	0.0167	139.06	<0.0001
X_1_ X_3_	0.0015	1	0.0015	12.32	0.0099
X_2_ X_3_	0.0022	1	0.0022	18.62	0.0035
X_1_^2^	0.0050	1	0.0050	41.73	0.0003
X_2_^2^	0.0076	1	0.0076	63.02	<0.0001
X_3_^2^	0.0001	1	0.0001	0.7852	0.4050
Residual	0.0008	7	0.0001		
Lack of fit	0.0004	3	0.0001	1.54	0.3346
Pure error	0.0004	4	0.0001		
Cor.Total	0.6471	16			

## Data Availability

The data presented in this study are available on request from the corresponding author.

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
