# Peer review of "The Design and Experimental Validation of a Biomimetic Stubble-Cutting Device Inspired by a Leaf-Cutting Ant’s Mandibles"

_biomimetics, 2023, doi:10.3390/biomimetics8070555_

Round 1

Reviewer 1 Report

Comments and Suggestions for Authors

This is a nicely done piece of work that examines whether cutting blades inspired from leaf cutter ants can improve the efficiency of soil tillage. 

The study is reasonably well motivated, the methods appropriate (with some caveats-see below) and the results are well presented and conclusive. Most of my concerns are minor. They are listed below, and with extensive comments on the ms to support these points, with a few other suggestions noted.

Major concerns:

1) L32-33 Please provide some additional context in the introduction about the function of the stubble cutting devices

2) L150. Please provide additional details about how the blade profile was determined, including  how many individuals were measured to derive this curve.  If you used multiple individuals did you measure both mandibles? Do you know if the left and right mandibles are the same? How do you know that the mandibles you measured were not subjected to wear or use-dependent changes? What's not clear is how general the data is and whether it reflects individual variation

3) L188. what measurements were taken to determine the conditions were similar, and how? Given that the data was collected on many different trials, it's essential to present the data on the measurements of soil condition to show your ability to consistently create the same conditions, and that variation in conditions does not affect the results. 

4) L220. Same problem as in point 3; how do you know conditions are uniform?

5) L319. The usefulness of the model in describing the results. or prediction, is a function of the model r^2, and should be provided for the quadratic fits for torque and power

6) L348. what parameters are you using to make this statement? Please refer to/present. Same comment for the statement on Line369 below

7) L 468. one thing not mentioned is whether soil type affects these results, and how.  It might be worthwhile to discuss how general you think your findings are with respect to this issue, and why. 

L487. So one of the most interesting things about the success of the BSCD is that the scale of what's going on in the ant vs the human device is quite different. Changing scales like this can often radically effect the physical forces. Is this just an example where the scaling does not create problems, and why would this be? Is it scale invariant like Re?  Or are things working differently in the ant vs the blades?

Also-is the specific form of the teeth important? Would a serrated blade have the same effect? WHy or why not? I guess I am wondering whether the inspiration from looking at ants was to add curves to the blade or whether the form of the curve derived from the ant is important? Do all leaf cutters have the same form? Might this affect the performance in different soil types?

You might not be able to address all of these issues, and certainly it's hard to provide a definitive answer, but these issues may be worth thinking about. 

Comments on the Quality of English Language

Author Response

Dear reviewer, thank you for your careful review and constructive suggestions regarding our manuscript. The manuscript was carefully revised and point-by-point response was listed below. We hope that your comments have been addressed accurately. The revised manuscript was marked with yellow color and the responses were presented in blue text.

Reviewer 2 Report

Comments and Suggestions for Authors

On the whole I have no adverse comments on this paper, other than that it misses a significant number of questions that would improve interest and knowledge, and probably the final product. In the comments below, the first half refers to improvements you can make to the paper; the second half suggests how much better it could be. At present you have a machine that appears to work (though no estimates are made showing effects of fatigue or wear, or expected life-time). I doubt that you know how the machine actually works (fracture and cutting occur at the micro-level, although they are initiated at all levels of hierarchy). So you have no indicators for optimisation of the design other that the performance of the current version.

Detailed notes:

Modifications and changes advised:

47 Roots do not act like 'steel bars'. Their interaction with the soil must be far more intimate than a steel bar would be. The model is of a fibrous composite, where the soil is the matrix, and the roots are the fibres. Refer to the papers by Roland ENNOS for a more accurate and complete account of how roots work as anchors. Remember that the roots themselves are composite fibres.

71 Cryptotympana atrata is a scientific name and must be in italics.

84-5 All generic and specific names should be in italics; in this instance Atta and Acromyrmex (NOTE - spelling)

93-100 Omit these comments on 'superlative' performance. They are related to scaling effects rather than intrinsic properties of the system.

121-124 Omit these lines - they are the common attributes that classify the ants as insects.

164 This equation can describe almost anything. Far more informative would be comparison of the shapes of the teeth of a range of species of ant compared with the morphology of the plants they cut into, including scale effects such as the size of leaf vein and the size and arrangement of the teeth. 

-- The tables of numbers should be omitted and summarised in a single line; similarly the details of the mathematical formulae should be summarised - they are all of the same form. 

Questions you might have asked (perhaps next time??)

With reference to the mathematical and experimental tests, there are many modifications that could be made:

-- with respect to the teeth, if they are different sizes and shapes in the ant, why? The large terminal tooth probably initiates the crack and may well anchor the mandible so that the rest of the leaf is constrained against the teeth. Our recent work on the saw of sawflies shows that the shape of the teeth is adapted to the type of plant material they cut through, whether it be the species of plant or the particular tissue.

-- do the teeth actually cut, or do they shear the leaf fibres? What does the fractured surface of the leaf look like? Are the fibres cut or are they pulled away from the surrounding matrix with a fibrous fracture surface?

-- I suspect the ant's mandibles slide over each other as they close. What happens if you vary the lateral distance between the blades as they pass each other in your machine? I suspect there is relatively little cutting. Look at the fracture surface of the cut ends of the roots. 

-- you have copied the outline of the teeth in your machine. Is there any reason why you didn't try other profiles?

There are many more questions that may, or may not, identify important variables in the design and implementation of the cutter.

Author Response

Dear reviewer, thank you for your careful review and constructive suggestions regarding our manuscript. The manuscript was carefully revised and point-by-point response was listed below. We hope that your comments have been addressed accurately. The revised manuscript was marked with red color and the responses were presented in blue text.

Round 2

Reviewer 1 Report

Comments and Suggestions for Authors

Almost all the previous comments have been addressed, with two important omissions:

1) As noted previously, there is no documentation of the soil conditions in the field and it must be established that these conditions are relatively constant and any potential variation does not affect the results.

2) Line 360, 381. The justification is still not well explained and possibly wrong. Please indicate precisely what "changing trend" refers to. THat is, what is it about the slope or other aspects of the surface that is being used to make the inference. Note also that using F values to indicate importance of a factor is not appropriate. Any comments about the importance of a given factor would come from the r-squared values.

3) Please indicate that the bars are std dev for in each figure legend for figures12-15

Author Response

Dear reviewer, thank you for your careful review and constructive suggestions regarding our manuscript again. The manuscript was carefully revised and point-by-point response was listed below. We hope that your comments have been addressed accurately. The revised manuscript was marked with red color and the responses were presented in blue text.
